# Measurement of the cosmic optical background using the long range reconnaissance imager on New Horizons

Michael Zemcov[1,2], Poppy Immel[1], Chi Nguyen[1], Asantha Cooray[3], Carey M. Lisse[4] & Andrew R. Poppe[5]

The cosmic optical background is an important observable that constrains energy production in stars and more exotic physical processes in the universe, and provides a crucial cosmological benchmark against which to judge theories of structure formation. Measurement of the absolute brightness of this background is complicated by local foregrounds like the Earth's atmosphere and sunlight reflected from local interplanetary dust, and large discrepancies in the inferred brightness of the optical background have resulted. Observations from probes far from the Earth are not affected by these bright foregrounds. Here we analyse the data from the Long Range Reconnaissance Imager (LORRI) instrument on NASA's New Horizons mission acquired during cruise phase outside the orbit of Jupiter, and find a statistical upper limit on the optical background's brightness similar to the integrated light from galaxies. We conclude that a carefully performed survey with LORRI could yield uncertainties comparable to those from galaxy counting measurements.

[1] Center for Detectors, School of Physics and Astronomy, Rochester Institute of Technology, 1 Lomb Memorial Drive, Rochester, New York 14623, USA. [2] Astrophysics and Space Sciences Section, Jet Propulsion Laboratory (JPL), 4800 Oak Grove Drive, Pasadena, California 91109, USA. [3] Department of Physics & Astronomy, University of California, Irvine, California 92697, USA. [4] Planetary Exploration Group, Space Department, Johns Hopkins University Applied Physics Laboratory, 11100 Johns Hopkins Road, Laurel, Maryland 20723, USA. [5] Space Science Laboratory, University of California at Berkeley, Berkeley, California 94720, USA. Correspondence and requests for materials should be addressed to M.Z. (email: zemcov@cfd.rit.edu).

The cosmic optical background (COB) is the summed emission from all sources outside of our Milky Way galaxy emitted at wavelengths roughly corresponding to those visible with the human eye. It is a powerful diagnostic of the emission from known astrophysical processes in galaxies including stellar nucleosynthesis, mass accretion onto black holes and the gravitational collapse of stars[1–3]. A comparison of the COB intensity to the surface brightness arising from known galaxy populations can reveal the presence of diffuse backgrounds produced by more exotic phenomena such as the decay of particle species outside the standard model or light from objects outside of galaxies[4–6].

Direct photometric measurement of the COB has proven to be challenging. The earth's atmosphere is several orders of magnitude brighter than the COB, and accounting for the various relevant emission, absorption, and scattering effects is a daunting task. Sunlight scattered from interplanetary dust (IPD) particles in the Solar system, known as Zodiacal light when viewed from the earth, also produces a large foreground to direct measurement of the COB from vantage points in the inner Solar system. Though progress has been made in carefully accounting for the atmosphere and Zodiacal light in the optical[7,8] and into the near-IR[9–14], as it is typically $>100$ times brighter than the COB small errors in this accountancy propagate to large errors on the COB[15,16]. It is thus desirable to measure the COB from vantage points where the earth's atmosphere and the light from IPD are not appreciable components of the diffuse sky brightness, such as the outer parts of our Solar system[17]. Though many planetary probes have had optical-wavelength cameras, they are rarely designed with the demands of extragalactic astronomical observations in mind.

Two exceptions to this are the early NASA probes Pioneer 10 and 11, which were instrumented with imaging photopolarimeters (IPPs) that returned measurements of the sky brightness ranging from 1 to 5.3 a.u. (ref. 18). These data have been used to measure both the decrease in the IPD light with heliocentric distance[19], diffuse light from the Galaxy[20,21] and the brightness of the COB itself[22,23] using the two IPP bands spanning 390–500 and 600–720 nm. The Pioneer measurements remain the most stringent constraints of the COB[23], and have uncertainties dominated by errors associated with subtracting galactic components including the integrated light from stars (ISL) and diffuse galactic light (DGL).

NASA's New Horizons spacecraft[24] recently performed the first detailed reconnaissance of the Pluto–Charon system. It includes as part of its instrument package the Long Range Reconnaissance Imager[25–27] (LORRI), an optical camera with sensitivity over a broad 440–870 nm half-sensitivity passband. Importantly, rather than a scanning photometer like the IPP, LORRI is a Newtonian telescope with characteristics including excellent pointing stability, a 20.8 cm diameter Ritchey-Chrétien telescope, an $0°.3 \times 0°.3$ instantaneous field of view, $1'' \times 1''$ pixels, and (crucially) real-time dark current monitoring. The achieved point source sensitivity of LORRI is $V = 17$ in a 10 s exposure in $4 \times 4$ pixel on-chip 'rebinning' mode, making it a sensitive astronomical instrument. As a result of this sensitivity and angular resolution, much of the starlight that challenged the earlier Pioneer measurements can be resolved out in LORRI images, providing a relatively clean measurement of diffuse astrophysical emission.

In this paper, we use archival data from the New Horizons checkout and cruise phases to measure the COB from several vantage points in the Solar system. We correct for dark current in the detectors, mask bright stars from the images, assess the amplitude of residual starlight, sunlight from interplanetary dust and diffuse galactic light, and correct for galactic extinction to measure $\lambda I_\lambda^{\mathrm{COB}} = 4.7 \pm 7.3(\mathrm{stat.})\,^{+10.3}_{-11.6}(\mathrm{sys.})\,\mathrm{nW\,m^{-2}\,sr^{-1}}$, giving a $2\sigma$ statistical upper limit of $\lambda I_\lambda^{\mathrm{COB}} < 19.3\,\mathrm{nW\,m^{-2}\,sr^{-1}}$, which excludes some of the early results in the literature. This measurement is based on a very limited data set with characteristics that complicate astrophysical examination. We conclude that a carefully designed survey of the COB from LORRI beyond the orbit of Pluto has the potential of definitively measuring its surface brightness away from the complicating effects of the earth's local interplanetary dust cloud.

## Results

**Data set.** In this study we concentrate on the $4 \times 4$ pixel 'rebinned' LORRI exposures, for which on-chip summing has been used to improve the surface brightness sensitivity over the native-resolution data. This rebinning mode is particularly advantageous in the small signal regime where the read noise penalty is large. The rebinned images have spatial resolution of $4''.3$ over a full $256 \times 256$ (super-)pixel frame. We term magnitudes in this LORRI band $R_\mathrm{L}$, as it is close to (though much broader than) the Johnson-Cousins $R$-band at 640 nm (ref. 28); in fact the LORRI bandpass covers essentially all of astronomical $V$, $R$ and $I$ bands.

New Horizons was launched 19 January 2006. The path of New Horizons through the Solar system is summarized in Fig. 1. Approximately 90 days after launch, some 359 dark data sets were acquired by LORRI while approximately 1.9 a.u. from the sun. During this time, the LORRI dust cover was in place over the telescope aperture, providing close to optically dark conditions. The cover was ejected on 2006 August 29, and shortly thereafter a series of images of the open star cluster Messier 7 were acquired. From these data, the New Horizons team determined a preliminary photometric calibration, a full-width at half maximum (FWHM) pointing jitter of 0.45 pixels, and geometric distortion $<0.2$ pixels across the field of view[27]. Following this, a series of short exposure test images was acquired, and at the beginning of 2007 the first science image was taken of Callirrhoe, a small irregular moon of Jupiter, with a full 10 s exposure time. A series of images of Jupiter and its immediate environment were then acquired during an encounter, which we do not consider here. Closest approach to Jupiter occurred on 28 February 2007. Following the Jupiter encounter, New Horizons entered cruise phase. LORRI data were acquired on an approximately annual basis, and consisted of 10 s observations of distant Solar system objects. The current public archival data records end 20 July 2014, approximately a year before the Pluto encounter.

We cut the data using requirements on integration time, solar elongation and thermal dust emission, following which we are left with fields $1 - 4$ whose characteristics are summarized in Table 1.

**COB measurement.** The brightness in an arbitrary image of the astronomical sky acquired above the Earth's atmosphere $\lambda I_\lambda^{\mathrm{meas}}$ can be expressed as:

$$\lambda I_\lambda^{\mathrm{meas}} = \lambda I_\lambda^{\mathrm{IPD}} + \lambda I_\lambda^* + \lambda I_\lambda^{\mathrm{RS}} + \lambda I_\lambda^{\mathrm{DGL}} + \epsilon \lambda I_\lambda^{\mathrm{COB}} + \lambda I_\lambda^{\mathrm{inst}}, \quad (1)$$

where $\lambda I_\lambda^{\mathrm{IPD}}$ is the brightness associated with interplanetary dust, $\lambda I_\lambda^*$ is the brightness associated with resolved stars, $\lambda I_\lambda^{\mathrm{RS}}$ is the brightness associated with residual starlight from stars too faint to be detected individually or the faint wings of masked sources, $\lambda I_\lambda^{\mathrm{DGL}}$ is the brightness of the DGL, $\lambda I_\lambda^{\mathrm{COB}}$ is the brightness of the COB, $\epsilon$ is a factor accounting for absorption in galactic dust, and $I_\lambda^{\mathrm{inst}}$ is brightness associated with the instrument including all potential contributions to the measured zero-point offset. The major difficulty with COB measurements is that, with the exception of $\lambda I_\lambda^*$, each of these sources can have an isotropic component, which is problematic since the COB itself is isotropic. As a result, care must be taken to understand and correct for the

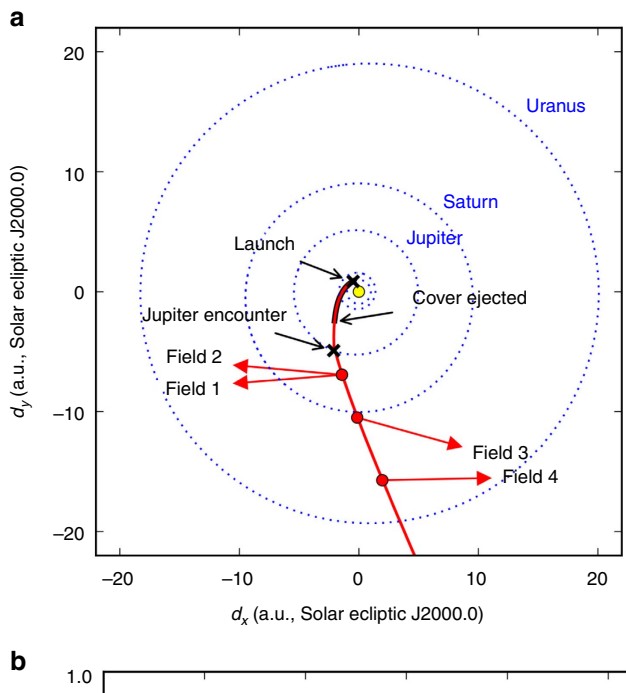

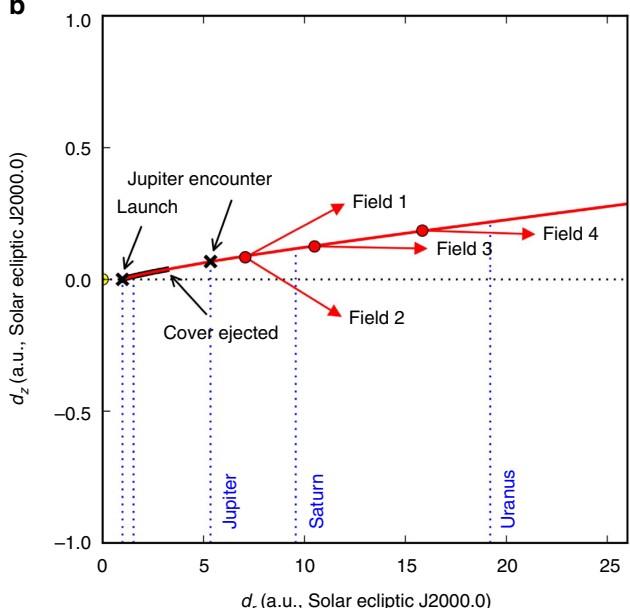

**Figure 1 | The trajectory of New Horizons through the solar system.** Data collection periods of relevance to this study are indicated. Both the $x - y$ and $r - z$ planes are shown (**a**,**b**, respectively), with the axes in solar ecliptic units and $d_r = \sqrt{d_x^2 + d_y^2}$. New Horizons was launched from Earth at 1 a.u., and the data with the LORRI dust cover in place were acquired at 1.9 a.u., just beyond Mars' orbit at 1.5 a.u. (inner blue dotted lines). The dust cover was ejected near 3.6 a.u., and the data were acquired before and during an encounter with Jupiter. The data considered here were taken between 2007 and 2010 while New Horizons was in cruise phase. The red vectors indicate the relative positions of fields 1 – 4 compared to the sun and plane of the ecliptic.

brightness of each component, particularly those that appear constant over angular scales similar to the field of view of the instrument.

We isolate $\lambda I_\lambda^{\mathrm{COB}}$ using three basic steps: mask stars near or brighter than the detection threshold to remove the effect of $\lambda I_\lambda^*$; subtract the diffuse components either originating in the instrument or from local astrophysical emission to isolate the

| Field number | $\alpha$ (J2000) hh:mm:ss | $\delta$ (J2000) hh:mm:ss | $\ell$ (°) | $b$ (°) | $e_b$ (°) | $A_V$ (mag) |
|---|---|---|---|---|---|---|
| 1 | 13:04:02 | 23:57:02 | 345.4147 | 85.7384 | 28.2096 | 0.06 |
| 2 | 10:47:36 | − 26:46:56 | 271.4532 | 28.4141 | − 31.5843 | 0.22 |
| 3 | 23:04:27 | − 7:07:00 | 66.2722 | − 57.6861 | − 1.0847 | 0.16 |
| 4 | 00:07:14 | − 1:15:00 | 98.8079 | − 62.0328 | − 1.8651 | 0.10 |

**Table 1 | Data sets used in this analysis.**

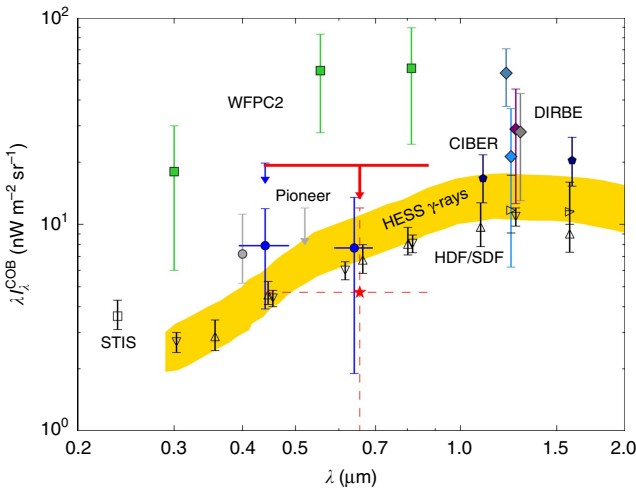

**Figure 2 | Measurements of the COB surface brightness.** The $\lambda I_\lambda^{\mathrm{COB}}$ determined in this study are shown as both an upper limit (red) and a mean (red star). We also show previous results in the literature, including direct contraints on the COB (filled symbols) and the IGL (open symbols). The plotted LORRI errors are purely statistical and are calculated from the observed variance in the mean of individual 10 s exposures; see Fig. 3 for an assessment of the systematic uncertainties in the measurement. We include the measurements from HST-WFPC2 (ref. 7; green squares), combinations of DIRBE and 2MASS[10-13] (diamonds; the wavelengths of these measurements have been shifted for clarity), a measurement using the 'dark cloud' method[8] (grey circles), and previous Pioneer 10/11 measurements[22,23] (blue upper limit leader and circles). The gold region indicates the H.E.S.S. constraints on the extragalactic background light[29]. We include the background inferred from CIBER[5] (pentagons). The IGL points are compiled from HST-STIS in the ultraviolet (UV)[62] (open square), and the Hubble Deep Field[63] (downward open triangles) the Subaru Deep Field[64,65] (upward open triangles and sideways pointing triangles) in the optical/near-IR. Where plotted, horizontal bars indicate the effective wavelength band of the measurement. Our new LORRI value from just 260 s of integration time is consistent with the previous Pioneer values.

diffuse residual component $\lambda I_\lambda^{\mathrm{resid}} = \epsilon \lambda I_\lambda^{\mathrm{COB}}$; and correct the mean residual intensity for the effects of galactic extinction to yield $\lambda I_\lambda^{\mathrm{COB}}$. Averaging over all the fields using inverse noise variance weighting, we determine that $\lambda I_\lambda^{\mathrm{COB}} = 4.7 \pm 7.3 \, \mathrm{nW \, m^{-2} \, sr^{-1}}$, where the uncertainty is purely statistical and is assessed from the scatter in the individual exposures. This gives a $2\sigma$ upper limit on the COB brightness of $\lambda I_\lambda^{\mathrm{COB}}(2\sigma) < 19.3 \, \mathrm{nW \, m^{-2} \, sr^{-1}}$. Our measurement and comparisons with previous measurements in the literature are shown in Fig. 2.

This measurement is also subject to various systematic uncertainties associated with the calibration and foreground removal. We carefully assess these errors by probing the allowed variation in each of the models and measurements to derive an overall calibration and systematic uncertainty budget,

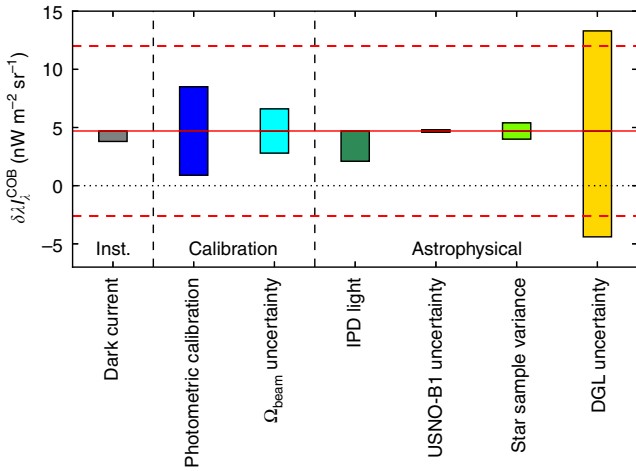

**Figure 3 | Summary of the various systematic errors in our determination of $\lambda I_\lambda^{COB}$.** The various sources of uncertainty are labelled, with the coloured bars showing their variation from the mean value we measure (red solid lines; see also Supplementary Table 3). Most of the errors are smaller than the statistical uncertainty of the measurement (dashed red lines), except for the uncertainty in the DGL model, which is large compared to the other errors. We do not show the errors associated with the optical ghosts and extinction correction as these are substantially less than a significant figure. The dominant uncertainties in this measurement are in fact not statistical, and to a great extent depend on the fields chosen and ancillary data available, so further observations in a dedicated survey program hold great promise.

summarized in Fig. 3. As the astrophysical errors identified in this analysis are uncorrelated, combining them in quadrature is an appropriate estimate of the total error present in the measurement. This is not the case for the calibration errors, which we add linearly and then sum in quadrature with the astrophysical foregrounds to give a conservative total systematic error estimate of $\{+10.3, -11.6\}\,\mathrm{nW\,m^{-2}\,sr^{-1}}$.

## Discussion

These data show the power of LORRI for precise, low-foreground measurements of the COB. The measurement presented here is not consistent with the earlier HST-WFPC2 constraints[7], but is consistent with both the Pioneer[23] and 'dark cloud[8]' measurements, as well as the $\gamma$-ray inference[29]. This measurement constrains the possibility of a COB significantly in excess of the expectation from IGL.

Though the bandwidth required to telemeter the data from the outer Solar system constrains the number of observations possible, with a carefully designed survey we should be able to produce a definitive measurement of the diffuse light in the local universe, and a tight constraint on the light from galaxies in the optical wavebands. LORRI's ability to resolve much of the starlight has significantly reduced the potential for foreground contamination compared to measurements from the Pioneer IPP, and the LORRI field is small enough that bespoke ground-based assessments of the faint starlight in each field are conceivable. As a result, a future LORRI survey would benefit from careful design and *pre facto* observations of the survey fields. Given the total integration time used in this measurement was only 260 s, a total integration time of $\sim 4.5\,\mathrm{h}$ would allow us to achieve $\sim 1\,\mathrm{nW\,m^{-2}\,sr^{-1}}$ statistical uncertainties. Because LORRI can allow 30 s integrations, this hypothetical measurement would require $\sim 500$ integrations, which is not prohibitive in terms of data storage nor telemetry requirements.

It would be particularly useful to observe high galactic latitude fields at a variety of ecliptic latitudes and solar elongations to search for IPD light. Though likely to be too faint to detect, models suggest there may be an increase in the IPD population towards the Kuiper Belt from collisional material[30]. This increase may be observable in the IPD light intensity with a carefully designed, deep survey. At the very least, observations of the inner Solar system from New Horizons' perspective may provide useful new information about the global structure of the IPD cloud. In addition, currently unpublished instrument calibration information such as susceptibility to off-axis light and detailed pointing stability assessments could improve the accuracy of this kind of measurement.

A primary lesson learned from this analysis is that, following the accurate removal of ISL, the DGL estimate becomes the largest source of uncertainty. Because of the uncertainty in the measured $100\,\mu\mathrm{m}$ background level, the DGL-$I_{100\,\mu\mathrm{m}}$ scaling, and the galactic latitude dependence of the scattering, this component varies by approximately the expected brightness of the IGL in each of our fields. For example, Field 1 is very close to the galactic pole where the DGL should be faint, but even here the models and observations suggest the DGL brightness could vary from 4 to $11\,\mathrm{nW\,m^{-2}\,sr^{-1}}$ at one s.d. Future observations with LORRI should concentrate on the lowest $I_{100\,\mu\mathrm{m}}$ fields available on the sky to minimize the uncertainty. If many statistically independent fields are sampled, the DGL-$I_{100\,\mu\mathrm{m}}$ linear regression technique we briefly explore should permit measurement of the optical-thermal infrared correlation precise enough to allow sub-$\mathrm{nW\,m^{-2}\,sr^{-1}}$ determination of the COB. Improved DGL characterization using other techniques are also of continuing importance.

This measurement of the COB brightness, while not currently as precise as those from Pioneer[23], is important as it suffers from completely different instrumental and foreground uncertainties as the existing measurement. It is also the only measurement sensitive to the $I$-band 700–900 nm wavelength range. Though some challenges remain, further data from LORRI could provide a definitive measurement of the extragalactic background light at optical wavelengths, and may be instrumental in completing our understanding of the history of stars and galaxies in the universe.

## Methods

**Observational data.** The basic New Horizons flight timeline is given in Supplementary Table 1, including a summary of the data taken during the checkout and cruise phases. The LORRI bandpass has an effective wavelength of $\lambda = 655$ nm for a flat-spectrum source, with half power response from 440–870 nm. Supplementary Table 1 indicates the dates of the observations, the notional targets, the number of integrations available, the exposure time per integration, as well as astrophysical information like the heliocentric distance $R_\odot$, solar elongation $\theta_\odot$, and the $100\,\mu\mathrm{m}$ specific intensity $I_{100\mu\mathrm{m}}$. In this work, we refer to each data set with a field number, running over D1–4, R1–10 and 1–4, using the numbering scheme presented below.

We performed cuts on the full data set to account for various factors affecting the data quality. First, we restricted our attention to the data with integration times $t_{int} > 1\,\mathrm{s}$, eliminating sets R1 − 4. Second, because the large-angle response of the LORRI telescope shows response from diffuse scattering of sunlight illuminating the light baffle[31], we remove data for which $\theta_e < 90°$. Finally, many of these LORRI data sets are taken at low galactic latitudes where the DGL is bright. We exclude data sets for which $I_{100\mu\mathrm{m}} > 10\,\mathrm{MJy\,sr^{-1}}$, which removes fields R5 and R7–10 from analysis. Though excluding much of the most useful data is not ideal, these fields are measured very close to the Galactic plane where the contamination from the local environment precludes careful measurement of the faint signals from either the COB or interplanetary dust in reflection.

**Data reduction.** The archival LORRI data are available in a format in which they have already been processed through several instrument calibration steps including bias correction, smear correction and relative pixel response correction[27,32]. In brief, the raw data consists of voltages measured at the end of the exposure reported in data numbers (DN). In the first step of the processing, the median of the dark reference pixels is used to subtract the global reference voltage, and a

reference 'super-bias' frame measured during ground testing is used to correct for bias variations over the array. Next, image smearing and flat field corrections measured during ground calibration are used to account for image smearing and relative pixel response. The files that are input to our processing have units data number per integration time and, though they have been partially processed, must be: astrometrically registered; masked for known detector defects, transients, and bright astronomical objects; corrected for instrumental effects; and calibrated to photometric units. Example images of the four science fields in this study are shown in Supplementary Fig. 1, calibrated to $\lambda I_\lambda$.

**Astrometric registration.** Astrometric registration is required to allow masking of bright stars, which is necessary to account for $\lambda I_\lambda^*$. Functionally, this means we want to ensure that each pixel in an image is accurately associated with a pair of right ascension and declination coordinates, $(\alpha, \delta)$. We determine the orientation and the scale of each image using the publicly available astrometric calibration software package http://astrometry.net[33]. The algorithm uses a four-step procedure in which: bright sources in each image are detected; the detected sources are divided into subsets whose relative positions are recorded and matched against a pre-built index; the solution is verified using predictive star position checks; and, the final alignment information is returned to the user in a Flexible Image Transport System (fits) header for each file. For the LORRI data, this algorithm successfully solved the astrometric registration for each field independently. Further, constant parameters of the instrument like the pixel size and image distortions are found to be consistent between observations over the entire data set, with very small uncertainties. The registration information returned by the software package is used to calculate $(\alpha, \delta)$ for every pixel in each image.

**Masking.** To reliably measure the diffuse sky brightness, it is necessary to exclude residual instrumental signal and detectable point sources from the images that contribute to $\lambda I_\lambda^{inst}$ and $\lambda I_\lambda^*$, respectively. We implement image masks to remove brightness associated with: stars near or brighter than the detection limit; pixels that may suffer from electronic or optical pathologies; and cosmic rays and other transient events.

To mask stars, we use the USNO-B1 catalog[34], which provides photometric fluxes in approximately Johnson-Cousins B, R and I bands over the entire sky. Though in some regions this catalog reaches completeness of $V = 21$, it is nonuniform and source fluxes are calibrated to only 0.25 mag accuracy. We synthesize the $R_L$ flux by fitting a linear model to the USNO-B1 measurements and compute the LORRI band-weighted integral of the flux for each source. To compromise between maximal removal of stars and minimal removal of galaxies that contribute to the COB, we mask the fields at a flux limit of $R_L = 17.75$, which is ∼1 mag below the $1\sigma$ point source sensitivity in the images. This threshold has the added benefit of moving the mask flux threshold away from the USNO-B1 catalog completeness limit where the survey uniformity is problematic. Given LORRI's small field of view, we calculate that the error introduced in the final COB estimates due to accidentally masking galaxies at the bright end of the number counts[35] is 0.006 nW m$^{-2}$ sr$^{-1}$. A search of the available optical data in these fields is consistent with the number counts, and we find that no exceptionally bright galaxies fall into these fields.

To build an appropriate source mask, we also require accurate knowledge of the instrument point spread function (PSF). We use a stacking method[36] to sum the emission from all $R_L < 16$ sources in the image to form an estimate of the LORRI PSF. Briefly, for each source brighter than the magnitude limit, we interpolate the image onto a ten times finer grid centered on the cataloged source position, and sum all such postage-stamp images. In each postage-stamp image, pixels far from any star images or masked pixels are used to calculate the zero point of the image. No lower magnitude limit is required as none of the images contain stars bright enough to induce non-linear response in the detector. The stacked PSF is shown in Supplementary Fig. 2. We compute the uncertainty in the PSF by performing two stacks, one stacking on a random half of the sources in the catalog and the other on the other half. We find a FWHM of $1.53 \pm 0.05''$, consistent with both laboratory[26] and in-flight[37] measurements of the PSF of $1.5''$.

We mask stars in each LORRI image by using the band-weighted magnitude estimate $m$ to calculate a radius around each source to exclude from analysis. This radius $r(m)$ is computed from:

$$r(m) = 2.5\left(\frac{m_{lim}}{m}\right)^2, \quad (2)$$

where the free parameters are determined empirically from the data. Here, $m_{lim} = 17.7$ at $R_L$-band and $r(m)$ has units of pixels. To assess the efficacy of this mask as a function of magnitude, we simulate noiseless images of the stars in each field per magnitude bin, apply the mask, and then calculate the residual surface brightness. We find the largest contribution to the residual brightness is from stars near the limiting magnitude, which contribute at most 0.15 pW m$^{-2}$ sr$^{-1}$ per source. Brighter sources have larger masks and contribute less total surface brightness as there are fewer of them. We calculate the total flux left in the images from residual unmasked star flux as part of the ISL assessment.

The data used in our COB study have a variety of Solar system objects as their primary targets. Though Haumea and Makemake should be faint in the optical ($R > 15$), even at a distance of 23.2 a.u. Neptune appears bright in the LORRI images ($R \sim 7$). To account for this, we uniformly mask the central $2'.3 \times 2'.3$ from each of the science images. In the case of Neptune, this corresponds to $35 R_{\text{♆}}$, which is significantly beyond the outermost known ring at $2.6 R_{\text{♆}}$ and the brightest moons (including Triton at $14.3 R_{\text{♆}}$). The Neptunian system does extend further, with a small moon orbiting $\sim 2,000 R_{\text{♆}}$ (at this distance $2°.2$) from the planet, and we cannot exclude the possibility that a dust halo far from the planet is reflecting sunlight and increasing the surface brightness in the LORRI image. We do not observe residual structure in these images, so any such contamination from a circum-Neptunian dust cloud is relatively faint. In principle, Haumea and Makemake may have their own dust clouds, and we cannot exclude the possibility from these data. As a result of these considerations, formally our measurements must be considered to be upper limits to the surface brightness of the COB. Future observations away from known Solar system objects would be beneficial in this regard.

For the Neptune field, the image of the planet is bright enough to induce charge transfer artifacts in the detector, so we mask three pixel wide stripes in both the vertical and horizontal directions of each image.

LORRI is known to have optical ghosting from reflections in the field flattening lens group for sources that fall between just off the field to up to $0°.37$ from the field[27]. This ghost is visible in the field 1 images, but not in the other fields. The central source mask removes a large fraction of this emission, but we also manually mask the ghosts in the field 1 images.

To reduce contamination from known defects, we mask both the outermost five pixels in the image, as well as pixels that are consistently non-responsive or saturated in the images. Finally, we apply a clip mask which excludes pixels $> 3\sigma$ from the mean value of each image. This excludes pixels with transient contaminants like electrical or digitization error and cosmic rays. On average, we exclude only ∼100 pixels in this $\sigma$-clipping step, corresponding to a 0.15% loss. In Supplementary Fig. 3 we show the same example images shown in Supplementary Fig. 1, but with the full image mask imposed.

**Dark current and reference pixel behaviour.** Since it cannot be separated from astronomically sourced photocurrent, an important potential contaminant in this measurement is the dark current of the detector, which contributes as an approximately isotropic component of $\lambda I_\lambda^{inst}$. The operating temperature of the LORRI charge coupled device (CCD) is $\sim 200$ K, so based on the performance of similar devices we might expect the dark current to be negligible. However, the COB measurement is more robust if the dark current contribution can be completely characterized.

An important feature of the LORRI CCD is the presence of rows of $4 \times 1,024$ (or $1 \times 256$ in rebinned mode) dark pixels. These are optically active pixels that are shielded from incident light by means of a metal lip, but are otherwise identical to the optically active pixels. These pixels are used in the LORRI data reduction pipeline to remove the combination of dark current and voltage bias in the images. In our study, these have the added benefit of giving a fixed reference of the detector array performance.

To characterize the long-term performance and stability of the detector, we compare the measurements performed with the dust cover in place against those taken with the dust cover off. The photocurrent in the reference pixels should be solely a function of the temperature of the detector[38], which is shown as a function of time in Supplementary Fig. 4. During post-launch operations before the cover was ejected, the detector system was passively cooled to its final operating temperature of $\sim 193$ K. As a result, the cover-on data were acquired at a significantly higher temperature than the optical data.

The detector manufacturer has empirically determined that, in equal-integration time exposures, the dark current at a temperature $T$ can be estimated from:

$$i(T) = 122\, i_0\, T^3 \exp(-6400/T) \quad (3)$$

with $i_0 \sim 10^4\, e^-\, s^{-1}$ per pixel. From this, we would predict $i(220\,\text{K}) \sim 2.8\, e^-\, s^{-1}$ per pixel and $i(193\,\text{K}) \sim 0.035\, e^-\, s^{-1}$ per pixel. In Supplementary Fig. 4 we show the mean of the 256 reference pixels in each of the four cover-on and four cover-off data sets. These show a decrease with temperature from a value of $\sim 545$ DN to $\sim 542$ DN. Assuming a model in which the bias voltage is a steady-state value where the dark current is negligible, we also plot both equation (3) and a free-amplitude fit of the same equation in the figure. Given the instrumental gain of 22 $e^-$ per DN, the free-amplitude fit gives a mean dark current of $7\, e^-\, s^{-1}$ per pixel at 220 K, which is a factor of 2.5 larger than the expectation but within the manufacturer's expected device to device variance. Assuming this factor, the expected dark current at 193 K is $0.09\, e^-\, s^{-1}$ per pixel.

The measured reference bias offset is subtracted from the images as part of the data processing pipeline. The value subtracted is the median of the 256 reference pixels[32], which is a reasonable estimate. However, on closer investigation we find that the median of the reference pixels can be biased due to the presence of large outlying pixels from, for example, cosmic ray hits. As a result, we measure a statistically significant correlation between the reference row median and the mean of the processed images. To correct for this bias, we instead use the $\sigma$-clipped mean of the reference row for reference subtraction. The mean is estimated by rejecting reference pixels with values $> 3\sigma$ after two iterations of rejection. Because in these data the median reference row value has already been removed, we correct the

mean value of the images by first adding back the reference row median and then subtracting the $\sigma$-clipped mean value. The correction is small, typically $<0.1$ DN per s. This procedure effectively removes the correlation between the subtracted reference value and the mean value of the processed image.

**Photometric calibration.** We calibrate the images from DN per s to Jy per pixel using aperture photometry. For each field, we identify two stars with flux $<1,000$ DN per s to avoid saturation effects, and greater than four pixels away from other sources or array artifacts. The pixel values are summed across a six pixel-wide aperture, and the background in a three pixel-wide ring three pixels away from the inner aperture excluding masked pixels is calculated and subtracted. The background-corrected aperture sum is then divided by the exposure time, giving the source flux $S$ in DN per s. Synthetic photometry is used to determine the magnitudes in the LORRI band using the USNO-B1 catalog. We calculate that the reference magnitude at $R_{L,0}$ given by the equation:

$$R_L = -2.5 \log S + R_{L,0} \qquad (4)$$

is $18.52 \pm 0.08$. The calibration factors are summarized in Supplementary Fig. 5.

The conversion from flux to surface brightness relies on both the frequency of the measurement $v_0$ and the measured solid angle of the PSF $\Omega_b$ through $\lambda I_\lambda = v F_v / \Omega_b$, where $\Omega_b$ is the instrument's 2D image-space impulse response function integrated over both dimensions. We estimate the effective frequency using the measured LORRI passband and assuming a flat input spectrum, which yields $v_0 = 458$ THz. $\Omega_b$ is calculated by summing the PSF shown in Supplementary Fig. 2 over the full $1'.8 \times 1'.8$ image. We find $\Omega_b = 2.64^{+0.18}_{-0.16}$ pixel$^2$, in good agreement with the FWHM = 1.5pix Gaussian model prediction of $\Omega_b = 2.54$ pixel$^2$.

We estimate the final surface brightness calibration factor to be $118 \pm 9$ μJy/(DN s$^{-1}$), which corresponds to $50.9 \pm 3.7$ nW m$^{-2}$ sr$^{-1}$ per DN per s. Following multiplication by this factor, the images are calibrated in surface brightness units and have associated masks that can be used to exclude pixels containing $R < 17.7$ point sources. The unmasked pixels in these images can be used to estimate the diffuse sky brightness. The raw diffuse sky brightness measurements corresponding to $\lambda I_\lambda^{\text{diffuse}} = \lambda I_\lambda^{\text{IPD}} + \lambda I_\lambda^{\text{RS}} + \lambda I_\lambda^{\text{DGL}} + \epsilon \lambda I_\lambda^{\text{COB}}$ are shown in Supplementary Fig. 6. To isolate the residual component of the observed emission associated with the COB, it is necessary to account for more local sources of emission, including residual interplanetary dust, residual starlight and diffuse galactic light. The contribution from each component is summarized in Supplementary Table 2.

**Interplanetary dust.** The population of $\sim 1$–1,000 μm dust particles in the Solar system reflects light from the sun and sources a diffuse sky brightness. Early in situ measurements of the dust distribution in the inner solar system from *Helios*, *Galileo*, *Ulysses* and *Pioneers* 8/9 show a sharp drop-off in the IPD density beyond 1 a.u., and confinement of the dust particles within 30° of the ecliptic plane[39]. It is difficult to formulate a mechanism that produces a long-lived population of dust out of the ecliptic plane[40], so the bulk of the IPD material is thought to reside at low inclination angles with respect to this plane and models of the dust distribution support this[30]. Interestingly, *Ulysses* measurements far above the ecliptic found a continuum level of particle events associated with the planar inflow of interstellar dust from the local hot bubble[39]. These dust particles are very small, with characteristic radii 1–10 nm, so do not effectively reflect sunlight at optical wavelengths. As a result, there is no expectation that IPD light is sourced far from the plane of the ecliptic.

In the outer Solar system, few in situ dust measurements exist. *Pioneers* 10 and 11 carried detectors that measured the flux of 5–10 μm particles[41]. *Pioneer* 10 reported data to 18 a.u. (ref. 42). *Pioneer* 11 made continuous measurements to $\sim 9$ a.u. and crossed the 3.7–5 a.u. region three times (once outbound and twice while transiting from Jupiter to Saturn), finding consistent results each time[43]. More recently, the Student Dust Counter (SDC) on New Horizons has measured the flux of 0.5–5 μm dust grains from 5 to 30 a.u. (refs 30,44,45). These measurements suggest an order of magnitude drop of the dust flux from 1 to 5 a.u., followed by a flattening of the particle flux to at least 20 a.u. Recently, a model has been generated that is consistent with all of the in situ measurements[30]; the predictions of this model scaled to a quantity that should follow the IPD light intensity are shown in Supplementary Fig. 7. From these in situ measurements, we would infer that the IPD population in the region over which the observations are performed is small and decreasing with distance.

In addition to in situ measurements, Pioneer 10 and 11 observed the optical-wavelength intensity of the background through the Solar system with a two-band imaging photopolarimeter[46]. The Pioneer 10 measurements from 2.4 to 3.2 a.u. exhibit a factor of $>25$ decrease in the surface brightness of two survey regions[19], both measured at $\theta_\odot > 102°$. The measurements beyond 3.25 a.u. are individually consistent with zero; averaging over these four measurements gives a $2\sigma$ upper limit on the IPD brightness of $<4.9$ nW m$^{-2}$ sr$^{-1}$, using the known conversion[47] between $S_{10}(V)$ and nW m$^{-2}$ sr$^{-1}$. The Pioneer 10 measurements are shown in Supplementary Fig. 7, and the upper limit on $\lambda I_\lambda^{\text{IPD}}$ is listed in Supplementary Table 2. No published analyses of IPD light in the Pioneer 11 data are available.

In the full set of 255 images, the LORRI data show no surface brightness change with heliocentric distance consistent with a variable contribution from the IPD, nor with viewing angle through the plane of the ecliptic.

**Residual starlight.** There are two contributions to $\lambda I_\lambda^{\text{RS}}$ in these data that from the unmasked wings of the PSF, and that from sources below the masking threshold. To calculate the residual starlight from the unmasked wings, we use the USNO-B1 catalog and measured LORRI PSF to simulate each flight image. These images are then masked with the flight mask, and the mean of the unmasked pixels is computed. We estimate that the residual starlight is negligible compared to the mean sky brightness in these images (see Supplementary Table 2).

The residual starlight from stars with $R_L > 17.7$ is challenging to calculate from real catalogs as they do not approach the required depth of $R \sim 25$. As a result, we use the TRILEGAL model to estimate the faint star flux, which models star fields as a function of position on the sky, photometric system, assumed stellar IMF, binary fraction, the sun's position and various parameters describing the Milky Way's thin disc, thick disc, halo and bulge[48]. The model returns catalogs of stars consistent with the observed number counts and known populations of stars. The number counts are returned to high precision, and TRILEGAL performs particularly well away from the galactic plane where the galaxy model is relatively simple. For each field's position, we generate ten TRILEGAL simulations of a $0°.3 \times 0°.3$ field corresponding to the LORRI image, complete to $R = 32$. For each simulation, we compute the mean surface brightness of the corresponding image. This results in the $\lambda I_\lambda^{\text{RS}}$ from faint sources listed in Supplementary Table 2. Even at the relatively faint masking threshold we apply, the residual flux from faint stars contributes a surface brightness comparable to the expected COB.

**Diffuse galactic light.** At optical wavelengths, dust in the galaxy reflects the local interstellar radiation field, and may also luminesce[49]. Similarly to the ecliptic dependence of light from the IPD, the DGL is brightest in the galactic plane and relatively faint at high galactic latitudes. Early Pioneer 10 measurements[50] found a factor $>10$ reduction between the DGL measured on the galactic plane and at the poles, and suggest a surface brightness of $\sim 150$ and $\sim 10$ nW m$^{-2}$ sr$^{-1}$, respectively. The implication is that nowhere on the sky can we ignore the contribution from the DGL. Since, on small scales, the spatial variation of the DGL[5] is fractions of a nW m$^{-2}$ sr$^{-1}$ in these LORRI data the primary effect is that of an overall surface brightness in the images. Due to LORRI's broad optical passband and the limited number of observed fields, with the LORRI data alone the DGL would be impossible to disentangle from the COB.

The dust grains responsible for the DGL are also heated by the interstellar radiation field (ISRF) and emit this energy thermally in the far-IR. As a result, the DGL is highly correlated with 100 μm emission in the optically thin limit where the optical photon scattering is simple[49]. Here we take advantage of this correlation and the excellent 100 μm all-sky surface brightness maps available[51,52] to estimate the contribution of the DGL to the optical surface brightness in each of the four fields. We have restricted our attention to high galactic latitude fields with $I_{100\mu m} < 10$ MJy sr$^{-1}$ in order to avoid optically thick dust as part of the data cut process. This allows us to take advantage of the linear relationship between thermal emission intensity and optical surface brightness.

We estimate the absolute surface brightness of the DGL in each field via the following relation:

$$\lambda I_\lambda^{\text{DGL}}(\lambda, \ell, b) = v\langle I_v(100\,\mu m)\rangle \cdot \bar{c}_\lambda \cdot d(b), \qquad (5)$$

where $v\langle I_v(100\,\mu m)\rangle$ is the 100 μm surface brightness averaged over the field, $\bar{c}_\lambda$ is the conversion from thermal emission intensity to optical surface brightness formulated below, and $d(b)$ is a function that accounts for the change in $c_\lambda$ due to scattering effects as a function of galactic latitude.

To estimate $v\langle I_v(100\,\mu m)\rangle$ we compute the Improved Reprocessing of the IRAS Survey (IRIS) 100 μm image[52] for each pixel in the LORRI images. We then subtract 0.8 MJy sr$^{-1}$ to account for the CIB brightness at 100 μm (refs 53,54), yielding the brightness of the dust in each field.

There are a variety of measurements of the scaling between the surface brightness in the optical/near-IR and at 100 μm, $c_\lambda = \lambda I_\lambda(\text{opt})/v I_v(100\,\mu m)$ (which is sometimes quoted as $b_\lambda = I_v(\text{opt})/I_v(100\,\mu m)$). We estimate $b_\lambda$ by fitting the mean 'ZDA04' model[55] to a compilation of measurements of $b_\lambda$ (ref. 49), which yields a best-fitting $c_\lambda$ and its uncertainty through multiplication by a factor of $10^{-6} \cdot (100\,\mu m/0.655\,\mu m)$. We then compute the wavelength-averaged value $\bar{c}_\lambda$ by integrating $c_\lambda$ weighted by the LORRI bandpass. This gives $\bar{c}_{655nm} = 0.49 \pm 0.13$.

Finally, we estimate $d(b)$ using the relation:

$$d(b) = d_0\left(1 - 1.1g\sqrt{\sin(|b|)}\right) \qquad (6)$$

where $d_0$ is a normalizing factor and the Henyey–Greenstein parameter $g$ is the asymmetry factor of the scattering phase function[56]. To estimate $g$, we compute the bandpass-weighted mean of an observation-constrained model for the high-latitude diffuse dust component of the DGL[57], and take the allowed variation in the mean from measurement and modelling errors as its uncertainty, yielding $g = 0.61 \pm 0.10$. To determine $d_0$, we normalize $d(b)$ at $b = 25°$ to be consistent with previous measurements[58]. As a check, we investigate the effect of using a larger estimate for $g$ consistent with scattering from dense molecular clouds below.

From this set of information, $\lambda I_\lambda^{\text{DGL}}$ can be computed. The value of $\lambda I_\lambda^{\text{DGL}}$ in each field is listed in Supplementary Table 2. Due to the relatively high galactic latitude and small size of the LORRI fields, and the large effective smoothing in the IRIS maps, we find it is unnecessary to account for the spatial variation in DGL in these fields.

As a check of the relatively uncertain direct DGL subtraction, we also compute a linear fit of the diffuse sky brightness minus the residual starlight against the 100 μm surface brightness in each of the four fields via:

$$\lambda I_\lambda^{\text{diffuse}} - \lambda I_\lambda^{\text{RS}} = \bar{c}'_\lambda \cdot v\langle I_v(100\,\mu m)\rangle + \lambda I_\lambda^{\text{resid}} \tag{7}$$

where $\bar{c}'_\lambda$ and $\lambda I_\lambda^{\text{resid}}$ are the parameters of the fit. We find a best-fitting $\bar{c}'_\lambda = 0.40 \pm 0.27$ and COB results consistent with the values determined via the direct subtraction method, but with much larger uncertainties since, in this fitting method, errors in the other DGL model parameters are folded into the measurement. When systematic errors are included, the estimates from both methods are similar in both absolute value and uncertainty, as expected.

**Extinction correction.** After accounting for the astrophysical foregrounds in the LORRI images, we retrieve the residual sky brightness measurements shown in Supplementary Fig. 8 and listed as $\lambda I_\lambda^{\text{resid}}$ in Supplementary Table 2. The per-field measurements are computed as the weighted mean of the individual exposures, where the weights are the inverse error on the mean in each 10 s exposure calculated from the variance of unmasked pixels. We take the uncertainty on the mean to be the s.d. of the individual exposures in each field.

To propagate these field averages to a measurement of the COB, it is necessary to correct for the effect of galactic extinction. One of the field averages is negative, for which an extinction correction is unphysical, so we choose to compute the mean residual brightness and then apply an equivalently generated extinction correction to that quantity. We first compute the uncertainty-weighted mean of the fields, and find $\lambda I_\lambda^{\text{resid}} = 4.5 \pm 6.9\,\text{nW m}^{-2}\,\text{sr}^{-1}$.

Next, we estimate the extinction correction by computing the mean of the two $A_R$ predicted by two models[51,59] in each field. We then compute the mean of the four field extinction measurements weighted by the same uncertainty weights used in the mean intensity computation. This yields an extinction correction of $A_R = 0.11$ mag. However, galactic extinction comprises two components, namely scattering and absorption. For point source observations, both of these remove light out of the line of sight, but since the COB is isotropic, the scattered light is replaced by light from other lines of sight in a conservative fashion. The proportion of these effects is roughly 40% absorption and 60% scattering, so our actual extinction correction is $A_R = 0.05$ mag. Applying this to $\lambda I_\lambda^{\text{resid}}$, we find $\lambda I_\lambda^{\text{COB}} = 4.7 \pm 7.3\,\text{nW m}^{-2}\,\text{sr}^{-1}$, where the errors are purely statistical. Extinction corrections do not apply to the systematic errors as they are all due to local mechanisms.

**Systematic and foreground error estimation.** The errors in this measurement can broadly be categorized as: statistical; systematics in the instrument, where we include the calibration uncertainty in this category; and systematics in the astrophysical foreground accountancy. The uncertainties in the measurement are summarized in Supplementary Table 3 and Supplementary Fig. 3, and their derivation is described in detail in this section. Mean values are always computed using the same field-to-field statistical weights as used in the average COB brightness calculation.

The statistical uncertainties are computed from the variance of a number of independent measurements, and as a result fold in a variety of sources of noise (detector, read out, photon, etc.) in an indistinguishable way. On the basis of a cursory inspection of the data and the known photocurrent, the noise is dominated by bit noise from the analog to digital converter, which suggests that in future measurements increased integration times would be beneficial.

As there are several steps in the data reduction, there are a corresponding number of potential errors in the instrumental corrections and data analysis we apply. First, the reference frame subtraction would have an error associated with it. However, because we later subtract the reference pixel values, we are removing a large part of the frame to frame variation that would lead to some offset error in this step. As a result, we fold all of the uncertainty associated with reference value/dark current into that step's error, as described below. Second, the application of a inter-pixel gain correction may have some intrinsic error, but because we perform photometric calibration after the flat field correction and we are not interested in the spatial structure of the images, errors in the applied pixel-to-pixel response should have a negligible effect on the final result. The only situation in which such an error could have a measurable impact is if the regions immediately surrounding the calibration stars had some local inter-pixel response different from the bulk of the detector array, and different than that measured during laboratory testing. To guard against this, we used calibration stars in random positions on the detector array and in multiple fields, and have shown the calibration to be consistent through the observation period. The photometric calibration uncertainty captures the remaining uncertainty from this effect.

As they are electrically identical to the photo-responsive pixels, and share the same read out chain, we have no expectation of or evidence for the reference pixel subtraction leading to any misestimation of the overall array offset. Extra variance in the final image brightness from the intrinsic measurement error of the 256 reference pixels that varies frame to frame is naturally accounted in the statistical uncertainty estimate. However, as the reference pixels are shaded from incident photons by a metal slat, it is possible that there is a <20% reduction in the dark current due to electromagnetic coupling to the shade (A. Reinheimer, private communication). To estimate the uncertainty from this effect, we compute 20% of the expected dark current in the pixels, which would cause a spurious image offset of $0.9\,\text{nW m}^{-2}\,\text{sr}^{-1}$. Because this would be in the direction of an under-subtraction, this error is in the positive-going direction and should be applied uniformly through the observations.

Optical ghosts were identified to be present in the central $r < 50$ pixel region of the LORRI images during laboratory testing[27]. The position and brightness of these ghosts depend on bright stars slightly off the imaged field, and they may be fainter than can be easily detected in the images. Though we masked these ghosts from our science images, ghosts fainter than the surface brightness limit to which we masked may be present. As a check of our masking procedure, we mask the full $r < 50$ region from the science images and recompute the resulting COB brightness through the entire analysis pipeline. When this augmented mask is applied to the images, we find a modest change to $\lambda I_\lambda^{\text{inst}}$ of $-0.1\,\text{nW m}^{-2}\,\text{sr}^{-1}$, that is, in the sense of an increased $\lambda I_\lambda^{\text{resid}}$. Unmasked optical ghosts would have the opposite sign, so we conclude there is no evidence for excess surface brightness from optical ghosts in these data.

On the basis of the dispersion of the aperture photometry measurements, we estimate the raw photometric calibration uncertainty of this measurement to be $\pm 7.3\%$. This compares well with the 0.25 mag catalog s.d. quoted as the per source photometric accuracy of the USNO-B1 catalog, which for 8 objects would give a photometric accuracy of 7.3%. USNO-B1 was ultimately calibrated from the Tycho-2 catalog, which itself has been calibrated to an accuracy of $\pm 2\%$ (ref. 60). We therefore estimate the absolute photometric uncertainty of this study to be $\pm 8\%$ of $\lambda I_\lambda^{\text{diffuse}}$, which corresponds to $\pm 3.8\,\text{nW m}^{-2}\,\text{sr}^{-1}$ on $\lambda I_\lambda^{\text{COB}}$.

The uncertainty in the solid angle of the beam also plays a role in the ultimate calibration uncertainty of this study. On the basis of half-half PSF stacking jack knife tests, we estimate the solid angle of the LORRI beam is known to $\pm 4\%$. This propagates to a 4% uncertainty on $\lambda I_\lambda^{\text{diffuse}}$, which corresponds to $\pm 1.9\,\text{nW m}^{-2}\,\text{sr}^{-1}$ on $\lambda I_\lambda^{\text{COB}}$. This uncertainty also includes the error in $\lambda I_\lambda^* + \lambda I_\lambda^{\text{RS}}$ due to our imperfect knowledge of $\Omega_{\text{beam}}$ on the conversion from flux to surface brightness.

The astrophysical foregrounds present in this study all have errors in their estimation. We have argued for a low level of IPD light in the outer Solar system, but explicitly quote the full $1\sigma$ uncertainty on the $R_\odot > 3.3$ a.u. Pioneer 10 measurements as an upper limit on the total IPD light contribution.

To account for the effect of the USNO-B1 photometric calibration uncertainty, we compute the estimate for $\lambda I_\lambda^{\text{diffuse}}$ after randomizing the reported magnitude of each source from the catalog by $\pm 0.25$ mag using a random Gaussian deviation per source. This has the effect of modifying the mask radius to be either inappropriately small or large, depending on the sign of the randomization. Over many masked sources, this should adequately probe the photometric error from the catalog calibration. On the basis of this calculation, we estimate the error to be $\pm 0.1\,\text{nW m}^{-2}\,\text{sr}^{-1}$ on $\lambda I_\lambda^{\text{RS}}$, resulting in an error on the COB of $\pm 0.1\,\text{nW m}^{-2}\,\text{sr}^{-1}$.

To calculate the variation in $\lambda I_\lambda^{\text{RS}}$ due to sample variance of the faint stars in our fields, we calculate the variance of ten realizations of the TRILEGAL model for each field, complete to $R = 32$. As these are relatively high galactic latitude fields, we find the s.d. in $\lambda I_\lambda^{\text{RS}}$ is $0.6\,\text{nW m}^{-2}\,\text{sr}^{-1}$ over the set, corresponding to $\pm 0.7\,\text{nW m}^{-2}\,\text{sr}^{-1}$ on $\lambda I_\lambda^{\text{COB}}$.

We compute the error in our estimate for the DGL brightness by propagating the error on the parameters in the various input functions. Specifically, we use $v I_v(100\,\mu m) = 0.80 \pm 0.25$ (refs 53,54), $g = 0.61 \pm 0.10$ (ref. 61) and $\bar{c}_\lambda = 0.49 \pm 0.13$ (ref. 49) as being consistent with the existing measurements. Propagating these errors through the DGL model gives $\sigma(\lambda I_\lambda^{\text{DGL}}) = \{-8.7, +8.2\}\,\text{nW m}^{-2}\,\text{sr}^{-1}$, resulting in an uncertainty of $\{-9.1, +8.6\}\,\text{nW m}^{-2}\,\text{sr}^{-1}$ on $\lambda I_\lambda^{\text{COB}}$. As a check of the effect of the relatively uncertain value of $g$, we use a compilation of results based on measurements of dense molecular clouds[61] that have a different mean scattering asymmetry factor $g = 0.75 \pm 0.1$ and find $\lambda I_\lambda^{\text{COB}} = 3.8\,\text{nW m}^{-2}\,\text{sr}^{-1}$, well within the quoted systematic uncertainty.

Finally, the galactic extinction correction is based on models that themselves have uncertainties. To bracket these, we take the allowed difference between the two extinction models[51,59] as our best estimate for $\sigma_{A_R}$. Over the four science fields, we compute the uncertainty-weighted allowable variation in the extinction correction to be a factor of 0.01 in surface brightness, which corresponds to a negligible error in $\lambda I_\lambda^{\text{COB}}$.

**Data availability.** The data that support the findings of this study are available from the NASA Planetary Data System at http://pds-smallbodies.astro.umd.edu/data_sb/missions/newhorizons/index.shtml.

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

## Acknowledgements

We thank both the New Horizons science team and the LORRI instrument team for their decades of dedicated effort designing, building and flying such a complex mission. Many thanks to H. Weaver and M. Richmond for useful discussions during the course of this work, and to the referees whose depth of knowledge and insights significantly improved the work. The New Horizons launch, Jupiter fly-by, and cruise phase data sets were obtained from the Planetary Data System (PDS). A.R.P. gratefully acknowledges NASA Planetary Atmospheres grant #NNX13AG55G.

## Author contributions

M.Z. developed the analysis and systematic error assessment pipeline, performed the foreground analysis and wrote the first draft of the paper. P.I. generated the data cuts and determined the photometric calibration of the instrument. C.N. collected the data from the archive and performed a variety of data quality checks. A.C., C.M.L. and A.R.P. worked on various aspects of foreground analysis and provided input on the low level analysis and workings of the instrument. All coauthors provided feedback and comments on the paper.

## Additional information

**Competing interests:** The authors declare no competing financial interests.

**Publisher's note**: 

