## [Peer Review File · Nature Communications]

Reviewer #1 (Remarks to the Author):

Referee Report

=====

Nature Communications manuscript NCOMMS-16-23185

Title: "Measurement of the Cosmic Optical Background using the Long Range Reconnaissance Imager on New Horizons"

Authors: M. Zemcov, P. Immel, C. Nguyen, A. Cooray, C. M. Lisse, & A. Poppet

Summary

The measurement of the extragalactic background light (EBL) at optical wavelengths represents an important observational cosmological constraint for current models of the universe ( Olbers' paradox). It is also one of the more challenging astronomical measurements anyone can attempt to carry out.

In this manuscript, the authors present two significant advances:

1. A careful analysis of the composition of the integrated light contributions in four high-galactic latitude fields observed with the Long Range Reconnaissance Imager (LORRI) on the New Horizons spacecraft, resulting in a new, significant, upper limit to the EBL in the wavelength range $\sim 400 - 900$ nm; in particular, their new 2σ -upper limit to the EBL is significantly below the earlier claimed "first detection" of the optical EBL by Bernstein et al. (2002), thus confirming the critical analysis by Mattila (2003).

2. A convincing demonstration that a definitive measurement of the EBL with LORRI is feasible with a potential commitment to observe a larger number of carefully selected, low-foreground fields, while New Horizons is still in communication range with Earth.

The paper is well organized and describes the instrument characteristics, the observations, and the methodology for removing all foreground emission sources in explicit detail. The special advantages of LORRI, i.e. the high angular resolution, allowing the masking of individual point sources down to faint magnitudes ($R \sim 17$), and the large distance from the Sun, eliminating all significant contributions due to scattering by zodiacal dust, are appropriately highlighted. This paper represents an important contribution to the EBL subject, and I recommend its publication following only minor corrections, as detailed below.

Specific Comments

1. Sect. 5., first paragraph:

It appears that the authors corrected the final measured EBL intensity for extinction by galactic foreground dust ($AR = 0.11$) as one would correct the flux from a point source. Extinction at optical wavelengths is a combination of absorption ($\sim 40\%$) and scattering ($\sim 60\%$). Both processes remove light from the line-of-sight to a point source, resulting in the combined extinction. A diffuse isotropic source such as the DGL, on the other hand, will be affected only by the absorption component, while light lost from the beam by scattering is fully replaced by scattered light into the line of sight from adjacent directions. Thus, the effective reduction of the EBL would amount to only ~ 0.04 mag for this case. This would change the value of $\lambda I_{\lambda cob} = 5.8 \pm 7.7$ nW m⁻² sr⁻¹ to $\lambda I_{\lambda cob} = 5.5 \pm 7.7$ nW m⁻² sr⁻¹.

While this change may be insignificant in light of the current uncertainties, it may become important at a future stage when the uncertainties can be reduced.

2. Sect 4.3.

Both observations and models yield ratios between optical/near-IR surface brightness of the DGI and the corresponding $100 \mu\text{m}$ background intensity with significant amounts of scatter, which is the principal source of the uncertainty of the current EBL measurement. There are, however, good physical reasons for variations in this ratio, and adopting a single, constant, value, i.e. $b_{655\text{nm}} = 0.33 \pm 0.07$ may not be the best approach.

For example, the far-IR intensity $I_{\nu}(100 \mu\text{m})$ per unit optical depth is determined by the Planck function $B(T_d)$, where T_d is the equilibrium dust temperature along the line of sight.

Schlegel et al. (1998; <http://irsa.ipac.caltech.edu/applications/DUST/>) provide an all-sky map of values of T_d . The four fields studied in the current investigation have values of $T_d = 17.82 \text{ K}$, 17.98 K , 18.21 K , and 17.71 K , respectively.

The ratio of the Planck function values for Field 3 and 4 is about 1.25.

It appears likely, therefore, that the value of the optical DGL intensity for Field 3 has been overestimated relative to the other three fields, which may be part of the reason why the residual intensity for Field 4 has a negative value after DGL subtraction.

Second, one should expect that $b_{655\text{nm}}$ exhibits a dependence on galactic latitude, especially at higher galactic latitudes where optically thin conditions prevail. The reason for this lies in the different coupling between the optical and far-IR background intensities with the galactic interstellar radiation field (ISRF). The far-IR intensity per unit optical depth depends on the dust temperature, which is determined by the integral over the ISRF over all solid angles. By contrast, the optical DGL per unit optical depth is strongly affected by the anisotropy of the ISRF, because the scattering phase function of interstellar grains is extremely strongly forward directed. Given that most of the ISRF arises in and near the galactic plane, one should expect that the DGL intensity shows a stronger gradient with increasing galactic latitude than the far-IR intensity over the range of latitudes where the optical depths for the DGL are within the optically thin regime. Applying these considerations to the data in the present investigation, it seems likely that the DGL intensity for Field 2, which has a significantly lower galactic latitude compared to the other fields, was underestimated by applying a fixed value of $b_{655\text{nm}}$. This may partly explain, why the residual intensity for Field 2 is highest among the four fields.

These considerations suggest to me that a more deliberate approach of estimating DGL intensities from measured far-IR intensities could be developed, which would result in a noticeable reduction in the uncertainties of the EBL measurements.

3. Two minor suggestions:

a.) Second but last sentence in the Abstract: I suggest changing the word order to: "...conversion from thermal emission intensity to optical surface brightness has large uncertainties."

b.) Page 3, first paragraph, last sentence: I suggest deleting the first "diffuse": "...errors associated with subtracting galactic components including the integrated light from stars (ISL) and diffuse galactic light (DGL)."

Reviewer #2 (Remarks to the Author):

This paper reports a new measurement of the cosmic optical background (COB) brightness, obtained with the LORRI instrument on board the New Horizons mission. While no secure COB detection is achieved, the resultant upper limit contains useful information to support or reject previous controversial measurements, and provides an independent constraint on the excess emission over the integrated light of known galaxies. At present, no other existing instrument can offer COB measurements at this accuracy level, which is only possible from the outer Solar system. Therefore I think this is an important contribution to the field of extragalactic astronomy, and would interest the broader community. The authors have successfully described their methods and limitations in considerable detail. I would recommend publication of this paper, once the following moderate/minor revisions are taken into account by the authors.

Abstract: The 2σ "statistical" upper limit of $21.2 \text{ nW/m}^2/\text{sr}$ is reported, but it is questionable

how meaningful this quantity is, given that the systematic error due to the calibration and foreground subtraction is roughly comparable to the statistical error. The net calibration and systematic errors, ± 4.2 and $\pm 4.6-26.3$, are never quoted in the text, so please clarify how these values are derived; I presume that the errors in Table 5 are summed in quadrature. Please also write out what "(stat.)", "(cal.)", and "(stat.)" mean here (the third one should be "(sys)")?

Sec 1, para 1, l.6: The expression "point source" is not appropriate, since it does not include extended galaxies.

Table 1: Please explain the color coding of the table cells.

Sec 2, para 3, l.3: What does "a V-band photometric offset of 18.94" mean? Is it the magnitude zero-point?

Sec 2, para 4: The authors found evidence for solar illumination of the instrument's light baffle and/or secondary mirror assembly in R6, whose solar elongation is 85.5 deg. I'm worried about this illumination also affecting the data sets 1 - 4, whose solar elongation (94.2 - 104.8 deg) is not much different from 85.5 deg.

Sec 3.2, para 2, the last sentence "We calculate that...": Please be more specific here. What is the surface brightness of the brightest galaxy expected to fall in the LORRI's field of view?

Sec 3.2, para 3: It is well known that PSF changes as a function of detector position and source brightness. The PSF uncertainty is currently estimated by comparing the two stacks generated from random subsamples, but this test should be performed with subsamples divided according to the above two quantities.

Sec 3.2, para 4: Please clarify the definition of m_{lim} and the unit of $r(m)$. Also, the sentence "This mask retains at most 0.15 pW/m/sr..." doesn't make sense to me; please rephrase or explain more.

Fig. 2: All the four images seem to have horizontal stripe patterns in the background counts. What are these? Do they affect the final error estimate of the COB brightness?

Sec 3.4, para 1: I think PSF photometry is better than aperture photometry for this calibration, since the latter is subject to the flux falling outside of the aperture. Is this aperture loss corrected?

Sec 3.4, para 2: Please clarify what is "the solid angle of the PSF", and how exactly it is used to convert flux to surface brightness.

Figs 7 & 9 captions: "The first two points have been offset slightly to improve clarity" -> "The first two points have been offset slightly in a horizontal direction to improve clarity".

Sec 4.3, para 1: Please provide the reference of the "early Pioneer 10 measurements".

Sec 5: The expressions "uncertainty-weighted" and "variance-wighted" are frequently used, but are misleading. I suspect the authors mean "inverse-variance weighted" with these words.

Sec 5, para 1: It would be more accurate to correct for the Galactic extinction in each of the four fields before computing the mean background value, rather than computing the mean background value first and then applying the mean Galactic extinction of the four fields.

Sec 5, para 1: Please describe here how exactly the quoted statistical error " ± 6.9 " (" ± 7.7 " after the Galactic extinction correction) is calculated. Is it derived by simply propagating the errors

of the four individual fields, or by calculating standard deviation of the values of the four fields, or does it include both of these?

Sec 5, para 2: The readers may be interested in the b_{λ} value estimated here.

Figure 10: Do the plotted error bars (and the upper limit line) include systematic errors? Do they represent 1sigma or 2sigma uncertainty? Please clarify.

Responses to Reviews of Nature Communications manuscript NCOMMS-16-23185

Title: "Measurement of the Cosmic Optical Background using the Long Range Reconnaissance Imager on New Horizons"

Authors: M. Zemcov, P. Immel, C. Nguyen, A. Cooray, C. M. Lisse, & A. Poppe

Replies to Reviewer #1

1. Sect. 5., first paragraph: *It appears that the authors corrected the final measured EBL intensity for extinction by galactic foreground dust ($AR = 0.11$) as one would correct the flux from a point source. Extinction at optical wavelengths is a combination of absorption (~40%) and scattering (~60%). Both processes remove light from the line-of-sight to a point source, resulting in the combined extinction. A diffuse isotropic source such as the DGL, on the other hand, will be affected only by the absorption component, while light lost from the beam by scattering is fully replaced by scattered light into the line of sight from adjacent directions. Thus, the effective reduction of the EBL would amount to only ~ 0.04 mag for this case. This would change the value of $\lambda\lambda_{cob} = 5.8 \pm 7.7$ nW m⁻² sr⁻¹ to $\lambda\lambda_{cob} = 5.5 \pm 7.7$ nW m⁻² sr⁻¹. While this change may be insignificant in light of the current uncertainties, it may become important at a future stage when the uncertainties can be reduced.*

This is an excellent point that we had missed entirely. We have added a new section to the Methods section that describes our new calculation that we hope adequately addresses this point. We have modified the COB amplitude and errors associated with extinction throughout the manuscript accordingly.

2. Sect 4.3.: *Both observations and models yield ratios between optical/near-IR surface brightness of the DGL and the corresponding 100 μ m background intensity with significant amounts of scatter, which is the principal source of the uncertainty of the current EBL measurement. There are, however, good physical reasons for variations in this ratio, and adopting a single, constant, value, i.e. $b_{655nm} = 0.33 \pm 0.07$ may not be the best approach.*

For example, the far-IR intensity $I_{\nu}(100 \mu\text{m})$ per unit optical depth is determined by the Planck function $B(T_d)$, where T_d is the equilibrium dust temperature along the line of sight. Schlegel et al. (1998; <http://irsa.ipac.caltech.edu/applications/DUST/>) provide an all-sky map of values of T_d . The four fields studied in the current investigation have values of $T_d = 17.82$ K, 17.98 K, 18.21 K, and 17.71 K, respectively. The ratio of the Planck function values for Field 3 and 4 is about 1.25. It appears likely, therefore, that the value of the optical DGL intensity for Field 3 has been overestimated relative to the other three fields, which may be part of the reason why the residual intensity for Field 4 has a negative value after DGL subtraction.

Second, one should expect that b_{655nm} exhibits a dependence on galactic latitude, especially at higher galactic latitudes where optically thin conditions prevail. The reason for this lies in the different coupling between the optical and far-IR background intensities with the galactic interstellar radiation field (ISRF). The far-IR intensity per unit optical depth depends on the dust temperature, which is determined by the integral over the ISRF over all solid angles. By contrast, the optical DGL per unit optical depth is strongly affected by the anisotropy of the ISRF, because the scattering phase function of interstellar grains is extremely strongly forward directed. Given that most of the ISRF arises in and near the galactic plane,

one should expect that the DGL intensity shows a stronger gradient with increasing galactic latitude than the far-IR intensity over the range of latitudes where the optical depths for the DGL are within the optically thin regime. Applying these considerations to the data in the present investigation, it seems likely that the DGL intensity for Field 2, which has a significantly lower galactic latitude compared to the other fields, was underestimated by applying a fixed value of $b_{655\text{nm}}$. This may partly explain, why the residual intensity for Field 2 is highest among the four fields.

These considerations suggest to me that a more deliberate approach of estimating DGL intensities from measured far-IR intensities could be developed, which would result in a noticeable reduction in the uncertainties of the EBL measurements.

These are also excellent points. To address them, we have modified the calculation of the DGL foreground to take into account: (i) the previous measurements of the DGL brightness; and (ii) the galactic latitude dependence of the brightness. Specifically, we have modified our DGL assessment to:

- (1) Fit the ZDA04 model spectrum to the existing set of b_{I} measurements (as collated in Ienaka et al 2013), resulting in a larger value of $b_{655\text{nm}}$ than before.
- (2) Used the results of Sano et al (2016) to account for the change in the DGL as a function of galactic latitude.
- (3) We now use a formal error assessment that takes into account the uncertainties in: (i) the brightness of the CIB at 100 μm ; (ii) the uncertainty in b_{I} based on the quoted scatter in measurements; and (iii) the uncertainty in g from the results in Sano et al.

Together, these modifications have had the result of slightly increasing the DGL amplitude, which has decreased the COB amplitude by $\sim 2 \text{ nW m}^{-2} \text{ sr}^{-1}$. However, this method also allows us to estimate our errors using all the available information, which has reduced them. We have heavily modified the description of the DGL estimation in the Methods section and hope this adequately addresses this point. We have modified the COB amplitude and errors associated with DGL throughout the manuscript accordingly.

To help clarify the presentation, we have also modified our previous use of the variable b_{I} to become c_{I} to bring our paper in line with the quantities defined in previous works. We explicitly give the conversion between b_{I} and c_{I} in the text.

3. Two minor suggestions:

a.) Second but last sentence in the Abstract: I suggest changing the word order to: “.....conversion from thermal emission intensity to optical surface brightness has large uncertainties.”

We had to remove the sentence to fit within the Nat. Comms. abstract length constraints, so this comment no longer obtains.

*b.) Page 3, first paragraph, last sentence: I suggest deleting the first “diffuse”:
“.....errors associated with subtracting galactic components including the integrated light from stars (ISL) and diffuse galactic light (DGL).”*

Yes, good catch.

Replies to Reviewer #2

Abstract: The 2sigma "statistical" upper limit of 21.2 nW/m²/sr is reported, but it is questionable how meaningful this quantity is, given that the systematic error due to the calibration and foreground subtraction is roughly comparable to the statistical error. The net calibration and systematic errors, +/- 4.2 and +4.6-26.3, are never quoted in the text, so please clarify how these values are derived; I presume that the errors in Table 5 are summed in quadrature. Please also write out what "(stat.)", "(cal.)", and "(stat.)" mean here (the third one should be "(sys)")?

In most physical disciplines upper limits are traditionally quoted as being purely statistical, at least when explicitly noted as such, and we choose to keep this convention. We agree that the calculation of the total systematic error was not adequately described, and in any case has changed with the new DGL assessment, so have added a new paragraph in the results section that describes this calculation in detail (it is not so simple as a straight quadrature sum). We believe the stat. and sys. nomenclature is standard in the field so retain the abbreviations, but have combined the calibration error into the systematic error for clarity.

Sec 1, para 1, l.6: The expression "point source" is not appropriate, since it does not include extended galaxies.

We used this phrase as we are used to considering instruments where COB galaxies would generally fall into a single spatial resolution element, but it is a fair point that this property varies from instrument to instrument. We have changed the sentence from:

"A comparison of the COB intensity to the surface brightness arising from known point source populations..."

to read:

"A comparison of the COB intensity to the surface brightness arising from known galaxy populations..."

Hopefully this is sufficiently broad to catch the various cases. We feel the qualifier "known" is necessary as people do occasionally discover faint/diffuse populations of galaxies which may have escaped detection in previous surveys.

Table 1: Please explain the color coding of the table cells.

Upon converting the paper from a generic format to that required for Nat.Comm., we found the table no longer worked well in the context of the introductory matter. Further, since the recent end of the Rosetta mission by crashing into comet 67P/Churyumov-Gerasimenko precludes the use of its camera for these type of observations, and the Dawn mission is slated to end in the next year, we felt this information did not add much to the discussion. We have therefore removed Table 1 and the sentence referring to it "Currently operating instruments capable of COB measurements well beyond 1 AU are few (see Table 1)." We hope you agree this information was not crucial to the discussion.

Sec 2, para 3, l.3: What does "a V-band photometric offset of 18.94" mean? Is it the magnitude zero-point?

Yes; we see this was stated in a confusing manner. Given the sentence's new position in the results section, we have changed the text to "a preliminary photometric calibration" and move the full discussion to the later methods section. We believe this is sufficient information to inform the casual reader of the sequence of events while not confusing the expert, who hopefully would look carefully at the methods section for detail.

Sec 2, para 4: The authors found evidence for solar illumination of the instrument's light baffle and/or secondary mirror assembly in R6, whose solar elongation is 85.5 deg. I'm worried about this illumination also affecting the data sets 1 - 4, whose solar elongation (94.2 - 104.8 deg) is not much different from 85.5 deg.

In response to this comment, we had a conversation with Andy Cheng (the PI of LORRI) about the design of the instrument. He provided us with a high-level schematic of the instrument and we discussed the off-axis response. The main point to be made here is that the secondary mirror assembly (plus support structure) is a full 150mm below the top of the optical baffle, and the baffle itself is flush with the top of the (flat) side of the spacecraft. As a result, sunlight cannot illuminate any part of the telescope or light baffle at solar elongations greater than 90 degrees, since the spacecraft blocks it completely. Essentially, it is physically impossible for images at solar elongations greater than 90 degrees to suffer from stray sunlight response.

As an interesting side note, Cheng et al (2010) [<http://spie.org/Publications/Proceedings/Paper/10.1117/12.859468>] calculate an expected flux of 10 DN/s/pixel at Pluto with a solar elongation of 30 degrees. Using their Figure 7 and referencing to 7.6 AU, we would expect an excess surface brightness of about 500 nW m⁻² sr⁻¹ in our images at a solar elongation of 85.5 degrees, which is consistent with our measurement. So it seems like the data are consistent in that regard. Again, let us emphasize that no response is expected at or beyond 90 degrees.

To clarify this issue in the text, we have modified the following:

"Second, because we did not have a precise measurement of the large-angle response of the LORRI telescope, we remove data for which $\theta_{\odot} < 90^{\circ}$ (for example, we found an anomalously large photocurrent and bright image of the secondary mirror in data set R6, which is likely due to the sun directly illuminating LORRI's light baffle and/or secondary mirror assembly)."

to read:

"Second, because the large-angle response of the LORRI telescope shows response from diffuse scattering of sunlight illuminating the light baffle\cite{Cheng2010}, we remove data for which $\theta_{\odot} < 90^{\circ}$."

Sec 3.2, para 2, the last sentence "We calculate that...": Please be more specific here. What is the surface brightness of the brightest galaxy expected to fall in the LORRI's field of view?

Based on the R-band number counts for galaxies, we expect between 2 and 3 galaxies brighter than the flux threshold to fall into a given field. This is supported by a NED search of the specific field positions and geometries, which (though ancillary optical observations are only available in 2 of the 4 fields) show two galaxies brighter than $R \sim 17.7$ in each. Working through the numbers, a mag 17 R-band galaxy has a flux of 486 μ Jy, which corresponds to 210 nW m⁻² sr⁻¹. Accounting for the averaging over the field, this

corresponds to a total error of $0.006 \text{ nW m}^{-2} \text{ sr}^{-1}$ on the resulting COB measurement. We would expect this, since the galaxies contributing the bulk of the COB are actually both much fainter and more populous, typically in the range 21-24 mag.

To clarify this issue, we have changed this last sentence to read:

“Given LORRI's small field of view, we calculate that the error introduced in the final COB estimates due to accidentally masking galaxies at the bright end of the number counts is $0.006 \text{ nW m}^{-2} \text{ sr}^{-1}$. A search of the available optical data in these fields is consistent with the number counts, and we find no exceptionally bright galaxies fall into these fields.”

Sec 3.2, para 3: It is well known that PSF changes as a function of detector position and source brightness. The PSF uncertainty is currently estimated by comparing the two stacks generated from random subsamples, but this test should be performed with subsamples divided according to the above two quantities.

Unfortunately, the quantity of available data precludes this type of jack knife test, since it would require not only a reasonable measurement of the solid angle of the PSF, but a detailed measurement of its two-dimensional shape. In a future measurement it would certainly be a desirable quantity to check.

We offer two pieces of evidence that, for the purposes of our analysis, our error estimate is sufficient:

- 1) For this COB measurement we only care about the *mean* quantities. Which is to say, if the PSF varies from position to position in the field, but we have a measurement of its mean, to convert from mean flux to mean surface brightness only requires knowledge of that mean solid angle and an assessment of the error in that knowledge. The stacking method, by construction, is determining exactly this quantity. Furthermore, the masking step is not very sensitive to the details of the PSF, since we intentionally stay in the “overmasked” regime.
- 2) The LORRI instrument team did not find any evidence for gross distortion over the field in their analysis of the check data. Which is not excluding the type of effect you point out, but is evidence that the optical performance of the telescope is not worse than their models predict.

Sec 3.2, para 4: Please clarify the definition of m_{lim} and the unit of $r(m)$. Also, the sentence "This mask retains at most 0.15 pW/m/sr ..." doesn't make sense to me; please rephrase or explain more.

To clarify, we modified this section to read:

“Here, $m_{\text{lim}} = 17.7$ at R_L-band and $r(m)$ has units of pixels. To assess the efficacy of this mask as a function of magnitude, we simulate noiseless images of the stars in each field per magnitude bin, apply the mask, and then calculate the residual surface brightness. We find the largest contribution to the residual brightness is from stars near the limiting magnitude, which contribute at most $0.15 \text{ pW m}^{-2} \text{ sr}^{-1}$ per source. Brighter sources have larger masks and contribute less total surface brightness as there are fewer of them.”

Fig. 2: All the four images seem to have horizontal stripe patterns in the background counts. What are these? Do they affect the final error estimate of the COB brightness?

These stripes are due to noise correlations in the output amplifier circuit, and are naturally present in any image of a sequentially read Si detector close to the noise floor. They are simply a manifestation of generic $1/f$ noise occurring in the read-out and amplification circuit on the time scale of a detector read.

The effect they have is to make the pixel variance an upper limit estimate of the Gaussian noise in the measurement. We were aware of this from the outset of the study so try to assess error via other metrics where possible (the variance in repeated measurements of the same quantity, for example). In general, the pixel variance will be an imperfect estimate of the absolute noise, but is a reasonable quantity to use for inverse-variance weighting where only relative changes in the level of the quantity matter.

To clarify, we have added to the relevant figure caption:

“The mild horizontal striping is due to noise correlations in the output amplifier circuit, and are naturally present in any image of a sequentially read Si detector close to the noise floor. This effect causes the pixel variance to be a poor estimate of the absolute statistical error, though it remains a reasonable weight for averaging calculations where only changes in the exposure-to-exposure variance are relevant.”

Sec 3.4, para 1: I think PSF photometry is better than aperture photometry for this calibration, since the latter is subject to the flux falling outside of the aperture. Is this aperture loss corrected?

We did use aperture photometry, and now having simulated the relevant processing have assessed that we have incurred a 0.02% error on the calibration factor as a result. This is a result of the large photometric aperture we use (which we can get away with as the images are far from crowded). Given the total calibration uncertainty, we think this is an acceptable bias factor. However, in future we will be careful to assess whether a more careful PSF fitting algorithm would provide benefits.

Sec 3.4, para 2: Please clarify what is "the solid angle of the PSF", and how exactly it is used to convert flux to surface brightness.

The solid angle of the PSF Ω is the equivalent angular area of the beam, i.e. the impulse response function integrated over both dimensions. It is used to convert to surface brightness through $I_{\lambda} = \nu I_{\nu} = F_{\nu} / \Omega$. To clarify, we have modified the sentence to read:

“The conversion from flux to surface brightness relies on both the frequency of the measurement $\nu_{\{0\}}$ and the measured solid angle of the PSF $\Omega_{\{b\}}$ through $I_{\lambda} = \nu F_{\nu} / \Omega_{\{b\}}$, where $\Omega_{\{b\}}$ is the instrument’s 2D image-space impulse response function integrated over both dimensions.”

Figs 7 & 9 captions: "The first two points have been offset slightly to improve clarity" -> "The first two points have been offset slightly in a horizontal direction to improve clarity".

Fixed.

Sec 4.3, para 1: Please provide the reference of the "early Pioneer 10 measurements".

This information comes from Gary Toller’s 1981 PhD thesis; a reference has been added.

Sec 5: The expressions "uncertainty-weighted" and "variance-wighted" are frequently used, but are misleading. I suspect the authors mean "inverse-variance weighted" with these words.

Yes, you're right, we had been loose with the language. We have modified instances of this phrase throughout to accurately reflect the calculations we perform. Please let us know if any ambiguities remain.

Sec 5, para 1: It would be more accurate to correct for the Galactic extinction in each of the four fields before computing the mean background value, rather than computing the mean background value first and then applying the mean Galactic extinction of the four fields.

It would, but we have the problem of having a negative point for which it is ambiguous how to correct the extinction. As a result, we compute the mean extinction correction using the same weights as is applied to the fields, which should ensure that each is weighted properly in the final mean.

Sec 5, para 1: Please describe here how exactly the quoted statistical error "+/-6.9" (" +/-7.7" after the Galactic extinction correction) is calculated. Is it derived by simply propagating the errors of the four individual fields, or by calculating standard deviation of the values of the four fields, or does it include both of these?

We have added a new paragraph toward the end of the results section describing this calculation in detail.

Sec 5, para 2: The readers may be interested in the b_{λ} value estimated here.

True - we have modified the sentence in question to read:

"We find a best-fitting $b_{\lambda} = 0.40 \pm 0.27$ and COB results consistent with the values determined via the direct subtraction method, but with much larger uncertainties since, in this fitting method, systematic errors in the DGL subtraction are folded into the parameter determination."

Figure 10: Do the plotted error bars (and the upper limit line) include systematic errors? Do they represent 1sigma or 2sigma uncertainty? Please clarify.

Yes, this was unclear. We have added to the figure caption:

"The plotted LORRI errors are purely statistical; see Figure 3 for an assessment of the systematic uncertainties in the measurement."

Reviewer #1 (Remarks to the Author):

Referee Report after First Revision

=====

Nature Communications manuscript NCOMMS-16-23185A

Title: "Measurement of the Cosmic Optical Background using the Long Range Reconnaissance Imager on New Horizons"

Authors: M. Zemcov, P. Immel, C. Nguyen, A. Cooray, C. M. Lisse, & A. Poppe

I apologize for the delay.

I have carefully examined the revised version of this paper. I want to commend the authors for their very serious effort of responding to my earlier comments, both in the substantial changes made in the analysis and in explaining these in their response to me.

Specific Comments:

As I had suspected, taking into account the galactic latitude dependence of the ratio of intensities of the optical/near-IR DGL to those of the far-IR dust emission did have a significant effect on the final value of the residual intensity of Field 2.

After revision, the DGL correction for this field increased by more than 80%, bringing the residual intensity down by about a factor 10(!), with a significant impact on the reported most likely value of the COB intensity. This further illustrates how uncertainties in the DGL will remain the principal source of uncertainty in measurements of the cosmic optical background.

This having been noted, it is of utmost importance that the latitude dependence of the DGL be approximated as reliably as possible.

For this reason, I would like to suggest that the authors re-examine the adopted value of the phase function asymmetry parameter $g = 0.75 \pm 0.10$, which they obtained from a display (Fig. 2) of a small sample of measured values of g in the paper by Sano et al. (2016). Instead, I suggest going back to an earlier, more comprehensive, summary of measured g -values published in Draine (2003, ApJ, 598, 1017, Fig. 10).

The principal physical parameter that determines the value of g is the ratio of the wavelength to the diameter of the particles that dominate the scattering at that wavelength. For a given fixed size distribution of grains, the value of g increases with decreasing wavelength, as is well illustrated by the measured g -values for reflection nebulae, shown in Draine's figure. We also know that grain sizes are different in different interstellar environments. All but one of the measured g -values displayed in the Sano et al. (2016) paper stem from Mattila (1970) and Witt et al. (1990). Both of these studies focused on dense molecular clouds, where grain sizes are typically larger due to conglomeration and formation of ice mantles. These environments generally present larger values of g than found in reflection nebulae, where radiation processing and heating are severe, or in the low-density diffuse ISM, where ice mantles are absent and conglomeration is highly inefficient. I suggest that the appropriate g -value for the analysis of the New Horizons data should be based on studies of the DGL, i.e. the top panel in Draine's figure. Given the broad spectral response of the New Horizons detector, my best guess of a suitable value would be $g = 0.62 \pm 0.10$, reflecting both the apparent wavelength dependence in the measured values as well as the theoretical models. This will moderate the impact of the analysis of Field 2 on the final result and will avoid objections from the interstellar dust community.

Aside from this single point, I fully support the publication of this manuscript in its current form.

Reviewer #2 (Remarks to the Author):

The authors have addressed all of my previous concerns, and also made a number of reasonable revisions following the comments from the other referee. I recommend the publication of the revised manuscript.

Responses to Reviews of Nature Communications manuscript NCOMMS-16-23185

Title: "Measurement of the Cosmic Optical Background using the Long Range Reconnaissance Imager on New Horizons", version 2.

Authors: M. Zemcov, P. Immel, C. Nguyen, A. Cooray, C. M. Lisse, & A. Poppe

Replies to Reviewer #1

1. *As I had suspected, taking into account the galactic latitude dependence of the ratio of intensities of the optical/near-IR DGL to those of the far-IR dust emission did have a significant effect on the final value of the residual intensity of Field 2. After revision, the DGL correction for this field increased by more than 80%, bringing the residual intensity down by about a factor 10(!), with a significant impact on the reported most likely value of the COB intensity. This further illustrates how uncertainties in the DGL will remain the principal source of uncertainty in measurements of the cosmic optical background.*

This having been noted, it is of utmost importance that the latitude dependence of the DGL be approximated as reliably as possible. For this reason, I would like to suggest that the authors re-examine the adopted value of the phase function asymmetry parameter $g = 0.75 \pm 0.10$, which they obtained from a display (Fig. 2) of a small sample of measured values of g in the paper by Sano et al. (2016). Instead, I suggest going back to an earlier, more comprehensive, summary of measured g -values published in Draine (2003, ApJ, 598, 1017, Fig. 10).

The principal physical parameter that determines the value of g is the ratio of the wavelength to the diameter of the particles that dominate the scattering at that wavelength. For a given fixed size distribution of grains, the value of g increases with decreasing wavelength, as is well illustrated by the measured g -values for reflection nebulae, shown in Draine's figure. We also know that grain sizes are different in different interstellar environments. All but one of the measured g -values displayed in the Sano et al. (2016) paper stem from Mattila (1970) and Witt et al. (1990). Both of these studies focused on dense molecular clouds, where grain sizes are typically larger due to conglomeration and formation of ice mantles. These environments generally present larger values of g than found in reflection nebulae, where radiation processing and heating are severe, or in the low-density diffuse ISM, where ice mantles are absent and conglomeration is highly inefficient. I suggest that the appropriate g -value for the analysis of the New Horizons data should be based on studies of the DGL, i.e. the top panel in Draine's figure. Given the broad spectral response of the New Horizons detector, my best guess of a suitable value would be $g = 0.62 \pm 0.10$, reflecting both the apparent wavelength dependence in the measured values as well as the theoretical models. This will moderate the impact of the analysis of Field 2 on the final result and will avoid objections from the interstellar dust community.

Thank you for your suggestion - the point that g values differ between the truly diffuse ISM and dense molecular clouds is certainly a reasonable one. We have studied Draine 2003 and have modified our approach to the calculation of g to match the one you outline above. Specifically, we determine a band-weighted value of $g=0.61 \pm 0.1$ (well done on getting so close by eye!) and a new normalization d_0 , and then propagate that through the pipeline to the estimate for the COB as before. We find the central value of the COB estimate changes by $0.9 \text{ nW m}^{-2} \text{ sr}^{-1}$, but perhaps more interestingly, the corresponding DGL uncertainty in the decreases somewhat.

We have propagated these changes throughout the paper, including Figures 1, 2 and S8 and the relevant tables. Within the text, we have modified the following:

“Finally, we estimate the $d(b)$ using the relation:

$$d(b) = d_0 (1 - 1.1 g \sqrt{\sin(|b|)})$$

where d_0 is a normalizing factor and g is the asymmetry factor of the phase function (Jura 1979). To compute g , we use a recent compilation of results (Sano et al 2016) to compute the bandpass-weighted mean of several individual measurements of g , and take the allowed variation in the mean from measurement errors as its uncertainty. This procedure gives $g = 0.75 \pm 0.10$. To determine d_0 , we normalize $d(b)$ to be unity at $b = 40^\circ$ to be consistent with the measurements of c_λ (Ienaka et al 2013).”

To read:

“Finally, we estimate $d(b)$ using the relation:

$$d(b) = d_0 (1 - 1.1 g \sqrt{\sin(|b|)})$$

where d_0 is a normalizing factor and the Henyey-Greenstein parameter g is the asymmetry factor of the scattering phase function (Jura 1979). To estimate g , we compute the bandpass-weighted mean of an observation-constrained model for the high-latitude diffuse dust component of the DGL (Draine 2003), and take the allowed variation in the mean from measurement and modeling errors as its uncertainty, yielding $g=0.61 \pm 0.10$. To determine d_0 , we normalize $d(b)$ at $b=25^\circ$ to be consistent with previous measurements (Lillie & Witt 1976). As a check, we investigate the effect of using a larger estimate for g consistent with scattering from dense molecular clouds below.”

In case we encounter a researcher with a different view to yours, we have retained a brief discussion of the result from the previous analysis in the systematic section; specifically, in the paragraph discussing the DGL uncertainty we have added:

“As a check of the effect of the relatively uncertain value of g , we use a compilation of results based on measurements of dense molecular clouds (Sano et al 2016) that have a different mean scattering asymmetry factor $g=0.75 \pm 0.1$ and find $\lambda_\lambda^{\text{COB}} = 3.8 \text{ nW m}^{-2} \text{ sr}^{-1}$, well within the quoted systematic uncertainty.”

Hopefully this covers the various cases sufficiently. We mark well the point that, although the precise value of g changes our result by a small fraction of an error bar, DGL contamination remains a dominant source of error. In fact, in having worked through this topic in detail, we noted a distinct lack of consensus regarding the various scalings which contribute to the DGL contamination, and hope we can interest the community in allowing us to perform further

measurements of these relations to constrain DGL to the $< 50\%$ level in these high latitude environments. Otherwise, the dream of determining the absolute brightness of the COB/CIB to the 10% level to be competitive with source counting measurements will remain just that.

Reviewer #1 provided confidential remarks to the editor, where they explain that their concerns have been fully addressed, that the authors have carried out the most careful analysis possible. They believe that publication of the paper will add a very valuable data point to this ongoing study, and share the authors' hope and expectation that additional observations with LORRI can be carried out.